# V-Dock: Fast Generation of Novel Drug-like Molecules Using Machine-Learning-Based Docking Score and Molecular Optimization

**DOI:** 10.3390/ijms222111635

**Published:** 2021-10-27

**Authors:** Jieun Choi, Juyong Lee

**Affiliations:** 1Department of Chemistry, Division of Chemistry and Biochemistry, Kangwon National University, Chuncheon 24341, Korea; jieun5074@gmail.com; 2Arontier Co., Seoul 06735, Korea

**Keywords:** protein-ligand docking, computer-aided drug discovery, docking score prediction, quantitative estimation of drug-likeness (QED), conformational space annealing (CSA), Lipinski’s rule of five, V-dock, MolFinder, deep learning, molecular property optimization

## Abstract

We propose a computational workflow to design novel drug-like molecules by combining the global optimization of molecular properties and protein-ligand docking with machine learning. However, most existing methods depend heavily on experimental data, and many targets do not have sufficient data to train reliable activity prediction models. To overcome this limitation, protein-ligand docking calculations must be performed using the limited data available. Such docking calculations during molecular generation require considerable computational time, preventing extensive exploration of the chemical space. To address this problem, we trained a machine-learning-based model that predicted the docking energy using SMILES to accelerate the molecular generation process. Docking scores could be accurately predicted using only a SMILES string. We combined this docking score prediction model with the global molecular property optimization approach, MolFinder, to find novel molecules exhibiting the desired properties with high values of predicted docking scores. We named this design approach V-dock. Using V-dock, we efficiently generated many novel molecules with high docking scores for a target protein, a similarity to the reference molecule, and desirable drug-like and bespoke properties, such as QED. The predicted docking scores of the generated molecules were verified by correlating them with the actual docking scores.

## 1. Introduction

Effective drug design requires optimizing various physicochemical properties, such as molecular weight, logP, number of hydrogen-bonded donor-acceptors, the polar surface area of molecules, and affinity with a target protein [1]. For efficient computational drug design, molecules that are similar to known active molecules but highly dissimilar to known molecules must be widely explored. However, obtaining biological data, such as binding affinity, of molecules via experimentation requires considerable time and resources. Protein-ligand docking calculations are commonly employed in the absence of binding affinity data. However, due to the large chemical space size, performing extensive chemical space search requires a prohibitively large number of computational resources. In this work, to overcome this limitation and facilitate the computational drug design process, we propose a new computational workflow. V-dock uses machine-learning algorithms to predict the docking score, followed by the application of the global molecular property optimization algorithm, MolFinder, to optimize the score [2]. We show that this new workflow significantly reduces the required computational resources compared to conventional approaches based on ligand docking and facilitates the design of novel drug-like molecules with the desired properties.

As machine learning (ML) algorithms have dramatically improved recently, various computational methods using reinforcement learning have been widely studied for designing novel drug-like molecules [3,4,5,6,7,8]. Gottipati et al. tried to enable molecular search in the chemical space by directly embedding synthetic knowledge in new drug designs [3]. To achieve this, they used a reinforcement learning (RL) model called policy gradient for forward synthesis to show that molecules are constrained to paths that can be synthesized and that all molecules proposed by the algorithm can be easily generated. Based on a graphical representation, the molecular deep Q-network (MolDQN) [4] algorithm uses the Markov decision process (MDP) [9] and RL to create molecules that maximize the selected properties. The MolDQN algorithm generates molecules that fit the purpose by adding, deleting, or modifying the covalent bonds of input molecules. Popova et al. suggested the ReLeaSE algorithm, which combines a stacked-RNN approach with RL to design novel inhibitor candidates for JAK2 [7]. The reward function that predicts the IC50 of a molecule is trained based on protein assay data. They showed that ReLeaSE could generate novel molecules with high IC50 values. Similarly, Lim et al. suggested a graph-based scaffold-constrained generative model using RL and predicted activity as a reward function [10].

Although these RL-based methods successfully generated novel molecules with accurate IC50 values, these RL-based models have a critical limitation. Most known RL-based generative models use the predicted activity as a reward function. To get reliable results, they require a large amount of experimental data to train an accurate reward function. However, very few targets contain known active compounds [11]. More than 99% of the protein activity bioassay data in the PubChem database [12] have been reported with fewer than 100 compounds, demonstrating the lack of data clearly. In addition, approximately 94% of the targets have less than ten active compounds, which prevents the training of an accurate activity prediction model due to the data imbalance. In other words, their dependency on experimental data makes the existing RL algorithms challenging to apply for novel drug targets.

To overcome this limitation, computational approaches that use ligand docking scores as approximations of experimental data have been suggested. Recently, virtual screening [13,14,15,16,17] algorithms combining ML algorithms with docking scores have been suggested. Gentile et al. [13] suggested a deep-docking approach that uses a docking score prediction model to reduce the resources required for docking computations. Preliminary docking calculations were conducted of a subset of ZINC15 [18]. Based on the docking results, a model that predicted the docking score of an unprocessed molecule was trained. Using this model, it was possible to increase screening enrichment by approximating the docking results of 13 million molecules of ZINC15 without performing actual docking calculations. Similarly, Berenger et al. suggested a lean-docking approach that used a linear support vector regressor trained with docking scores of 10,000 molecules [14]. These results showed that efficient massive virtual screening is possible using a docking score prediction model without deteriorating the screening power.

In addition to virtual screening, attempts have been made to generate molecules by directly optimizing a docking score [19,20,21,22]. Recently, Boitreaud et al. [23] suggested the OptiMol approach based on binding energy optimization for drug design using a generative model and docking using adaptive sampling (CbAS) [24] to maximize the objective function. They showed that the method finds compounds with high affinity for a given target protein more efficiently than reinforcement-based algorithms. Ma et al. suggested a structure-based de novo molecular generator approach, SBMolGen, by combining an RNN-based SMILES generator and a Monte Carlo tree search with docking simulations [25]. In the SBMolGen approach, a generated molecule is docked with a target protein, and the result is used to provide feedback to the generator to optimize the docking score. Jeon and Kim developed the MORLD method [20], which combines RL with docking using QVina2 [26]. An initial molecule was modified through RL to optimize the docking score, drug-likeness [27], and SA scores [28]. However, most docking-assisted molecular design approaches performed actual docking calculations to obtain docking scores during the optimization process. Performing docking calculations hinder extensive sampling in chemical space because it requires a significant amount of computational time. Additionally, to perform the docking of a generated molecule, the molecule’s 3D structure and partial charge need to be calculated, which is a bottleneck in the computational process. Therefore, due to the vast size of chemical space, a much faster evaluation of the docking score is essential for its extensive search.

This study proposes a V-dock approach to generate novel drug-like molecules efficiently by combining the global optimization approach with a docking score predictor for drug targets for which very little experimental data is available. We identified that the docking score could be accurately predicted using only the 1D and 2D descriptors. This significantly accelerates the molecular optimization process and enables extensive exploration of the chemical space. Thus, to find novel drug-like and readily synthesizable molecules efficiently, we devised an objective function to make molecules with higher objective values more drug-like. The function consists of three terms: predicted docking score, drug-likeness score (QED), and similarity to a reference molecule. The MolFinder [2] algorithm, the global molecular property optimization algorithm using SMILES, was used to optimize the given objective function. Previously, we demonstrated that MolFinder generates more optimized molecules than known RL-based methods. The benchmarks showed that MolFinder could generate novel molecules similar to a reference molecule with high QED, SA-score values.

Using the V-dock approach, we generated possible inhibitors of Werner syndrome ATP-dependent helicase (WRN) [29,30]. The number of known inhibitors of WRN is limited, making it a challenging target for existing reinforcement-based models, which heavily depend on large amounts of biological activity data. Compared to existing generative models, MolFinder has the advantage of generating diverse molecules with desirable properties without many experimental data because it samples novel molecules in a combinatorial manner in the SMILES space [2]. In addition, we identified that our approach, which uses a docking score predictor, accelerates the molecular optimization process 255.5 times compared to actual docking calculations. Finally, we show that the predicted docking scores of novel molecules designed via optimization correlate well with the actual docking results. This suggests that our V-docking approach is a promising and generalized approach for generating novel molecules that satisfy the specific properties of interest.

## 2. Results and Discussion

### 2.1. Docking Energy Prediction Model Training Result

Of the 100,000 molecules, 85,432 were docked, and the 14,568 for which docking calculations were not completed were larger than 30 Å, the size of the set grid box. As a result of the docking calculations for 85,432 molecules, the minimum value of docking energy calculated was −16.4 kcal/mol. The maximum value of docking energy calculated was −0.4 kcal/mol, and the average score was −7.47 kcal/mol (Figure 1a).

Using the docking energy prediction model, the Pearson correlation coefficient between the predicted values and the actual docking energy values of the test set was 0.94. A mean square error value of 0.34 kcal/mol (Figure 1b) was obtained. The Pearson correlation coefficient of the training set was 0.85, and that of the whole set was 0.89. These results showed that the docking energy could be predicted accurately using only the SMILES representation.

The average computational time taken for the actual docking calculation of a molecule was 15.21 s. Meanwhile, the average computational time of the docking energy predictor for a single molecule was 0.0595 s (Figure 2). Therefore, using the predictive model is 255.5 times faster than an actual docking calculation. This indicates that using the docking energy predictor enables molecular design via docking energy optimization using global optimization approaches or reinforcement algorithms with accessible computational time and reasonable accuracy.

### 2.2. Optimization Results for Generated Molecules

To optimize the weight parameters, we generated molecules via global optimization of the objective function of Equation 1 by using MolFinder with different weight parameters. The MolFinder calculations were repeated 100 times each using a weight parameter, $\omega_{D}$ = 0.1, 0.05, 0.03, and 0.01, and the properties of the generated molecules were investigated with the weight changes (Figure 3). To reduce computational complexcity, $\omega_{Q}$ and $\omega_{S}$ are fixed to the initial value, one. For the MolFinder calculations, the number of preferred molecules (seed) was set to 120, and 500 molecules were generated for each iteration. The distributions of the predicted docking energy, QED, and the similarity of the generated molecules to ML216 confirm that the targeted properties are optimized compared to those present in the existing SureChEMBL dataset.

Using the lowest weight of 0.01, molecules with higher (worse) docking energies than the 500 molecules having the lowest docking energy in the SureChEMBL datasets were obtained. The generated molecules are shown in Appendix A However, using larger weights, it was confirmed that all of them led to molecules with lower (better) docking energies than the existing molecules in SureChEMBL. Molecules designed with $\omega_{D}$ values of 0.1 and 0.05 had lower docking energies than those generated with an $\omega_{D}$ value of 0.03. However, QED values were similar to those generated with a $\omega_{D}$ value of 0.03, and the similarities to ML216 were relatively low, less than 0.4. In other words, among all the tested $\omega_{D}$ values, when the $\omega_{D}$ value was 0.03, molecules with low docking energy, high QED, and high similarity were obtained. These results indicated that if only docking energy was considered, it could be further lowered by increasing the value of the weight. However, because the purpose of this study is to generate a drug molecule that has biological activity on the desired target protein and satisfies various physical properties, only the $\omega_{D}$ value of 0.03 was used for subsequent calculations.

We generated 500 molecules with an ω_D_ value of 0.03 and compared their properties with those of the existing molecules in SureChEMBL. SureChEMBL was filtered using Lipinski’s rule to identify the molecules with the lowest docking energy. The ten molecules having the lowest docking energy among the filtered molecules are shown in Figure 4a. Forty additional molecules with the lowest docking energies are illustrated in Appendix A. The mean QED of the ten best molecules from SureChEMBL was lower than 0.53, and the mean of similarity was also low (less than 0.34). The ten best molecules with the lowest objective values among the finally generated molecules are shown in Figure 4b. Forty additional best molecules are illustrated in Appendix A. We found that the mean QED of the top ten generated molecules was 0.76, and the mean similarity was 0.83, which were much higher than the values for the best molecules from SureChEMBL.

The generated molecules have a structure that is partially similar to ML216, an inhibitor of BLM. None of the 500 molecules generated in the last iteration were found in SureChEMBL. They also did not exist within the entire ChEMBL dataset. These results confirm that all 500 molecules are novel and have desirable properties.

The distributions of the total objective function and QED values of 500 generated molecules with the highest similarity to the reference molecule were compared with those of SureChEMBL (Figure 5a). Given that the generated molecules had a low objective function distribution ranging from −1.7 to −1.8, it was confirmed that the molecules had been optimized for the desired properties. Meanwhile, SureChEMBL distributed objective function values between −0.6 and −1.5.

The distribution of the individual properties of the objective function is illustrated in Figure 5b. The distributions demonstrate that our workflow successfully optimized the three selected properties of interest simultaneously. The generated molecules have much higher docking scores, QED values, and more remarkable similarities to the reference than the known molecules. The docking score of ML216, a known inhibitor of BLM, was −8.5 kcal/mol. Our approach generated many molecules with better docking energies than ML216, indicating that these molecules can be potential leads for the development of WRN inhibitors.

Overall, molecules with higher QED values had worse docking scores. This indicated that it is difficult to discover molecules from the database satisfying two criteria simultaneously: a high QED score and a high docking score. The QED values of the generated molecules had a value between 0.6–1.0 and are significantly higher than that of SureChEMBL, which are mostly less than 0.6. Furthermore, the docking energies of the generated molecules were distributed between approximately −10 kcal/mol and −6 kcal/mol, while the docking energies of the SureChEMBL molecules were lower and more widely distributed. At the same time, the generated molecules had much higher similarities to the reference than the known molecules. Molecules with a similarity ≥ 0.5 with the reference molecule do not exist in the SureChEMBL dataset (85,161 molecules). This demonstrated that the generation of molecules depended on the objective function and not the input data. We also calculated the distributions of the QED and similarity of ChEMBL, which are displayed in Appendix A. In ChEMBL, only 97 out of 1,941,412 molecules had a similarity of more than 0.5 with the reference molecule. However, all molecules generated by V-Dock had higher than 0.8 similarities to ML216. In summary, we demonstrated that V-Dock successfully generated novel molecules that satisfied multiple criteria.

To further investigate the effect of the weights of the other terms, QED and a similarity to the reference, we performed additional calculations with different $\omega_{Q}$ values (Appendix A. When $\omega_{Q}$ is reduced to 0.5 from 1.0, the average QED of generated molecules reduced from 0.92 to 0.81~0.84. With $\omega_{Q}=0.1$, the average QED reduced significantly to 0.14. When $\omega_{Q}$ is increased to 5.0, the average QED increased to 0.944.

We also performed additional calculations with different weights for the similarity term, $\omega_{S}=0.7$ and 0.4 (Appendix A). When $\omega_{S}$ is reduced to 0.7, the average similarity slightly reduced from 0.58 to 0.51. The generated molecules are sharing similar fragments with ML216 (Appendix A). When $\omega_{S}$ is reduced further to 0.4, the average similarity significantly dropped to 0.28. The resulting molecules are highly different from the reference (Appendix A). These results show that the average of each term can be tuned by changing the weight parameters.

### 2.3. Molecular Validation Designed with V-Dock

We performed a re-docking computation using the generated molecules to verify whether the molecules designed with the V-dock method showed good docking scores (Figure 6, Appendix A. We performed docking simulations using 100 generated molecules with the lowest objective function values. The trained docking energy prediction model compared the predicted docking energy with the true docking energy value obtained, with a correlation coefficient value of 0.61 (Figure 6). This correlation confirmed that molecules generated via optimization of the predicted docking score had good docking energies, indicating that our method can generate promising drug candidates. In addition, these results showed that the docking energies of molecules could be estimated quickly using the trained predictor. In summary, we proved that the V-dock approach generates molecules with desirable drug-like properties and predicts them much faster and more accurately than docking calculations for such molecules that lack experimental information.

### 2.4. Possible Limitations of the V-Dock Approach

The fundamental assumption of V-dock is that the docking score of a novel molecule is correlated with its true binding affinity to a target protein. In other words, the efficiency of V-dock heavily depends on the accuracy of the docking score. Thus, one of the most critical limitations of V-dock is that the correlation between docking scores and experimental binding affinities varies by targets [31,32]. Although no existing docking score is perfectly correlated with all protein-ligand complexes in general, the large-scale benchmark by Wang et al. with 2002 protein-ligand complexes showed that the scoring function of Autodock Vina is correlated with experimental binding affinities reasonably well [33]. Autodock Vina achieved the best scoring power from the benchmark, the highest correlation between docking scores and experimental affinities, with an average Pearson correlation coefficient of 0.564, among all tested docking programs, including commercial programs. In summary, the Autodock Vina score is one of the best approximations available at present.

### 2.5. Drug-likeness of the Generated Molecule

To further confirm the drug-likeness of the generated molecules, the seven properties considered for QED (molecular weight, logP, number of hydrogen bond donors, number of rotatable bonds, number of aromatic rings, and surface area of molecules) were compared with SureChEMBL (Figure 7). The seven physicochemical properties of a given molecule are calculated using the RDKit library version 2019.09. The comparison showed that the molecules generated using the V-Dock method satisfy Lipinski’s rule of five better than those present in SureChEMBL. For the molecules generated using the V-Dock method, there were no molecules with a molecular weight of 500 Da or more and a logP of 5 or higher. The numbers of H-bond acceptors and donors were not more than ten and five, respectively. Finally, the number of rotatable bonds, the number of aromatic rings, and the total polar surface area were found to be 10, 5, and 140 Å^2^ or less, respectively. Therefore, we confirmed that the physical properties of the generated molecules had favorable conditions for drug-likeness. For molecules present in SureChEMBL, the molecular weights were widely distributed, and there were molecules with molecular weights higher than 500 Da and many molecules with logP values greater than 5. In these molecules, five other properties were also widely distributed without satisfying Lipinski’s rule of five.

## 3. Materials and Methods

### 3.1. Overall Workflow of V-Dock

The workflow of this study is shown in Figure 8. As the first step, we randomly selected 100,000 molecules from the SureChEMBL database. Docking calculations were performed using this subset, and these data were used to train a docking energy prediction model based on SMILES. Using this predictive model and the molecular sampling algorithm, MolFinder, we designed molecules via global optimization of the desired physicochemical properties. Finally, the predicted docking energies of the generated molecules were confirmed using docking calculations. Through this, we demonstrated that with the assistance of predicted docking scores, novel molecules with high docking energies and desirable physical properties could be quickly generated using only SMILES, even with limited experimental data.

### 3.2. Molecular Database

Molecules from the SureChEMBL dataset retrieved on 20 August 2020, were used to train predictive models for predicting docking energy and carrying out docking computations [34]. The SureChEMBL database consists of publicly patented molecules related to biological targets. There were 20,227,433 molecules in the database, of which we used the SMILES representation of 100,000 randomly extracted molecules. SMILES strings in this dataset were used for protein-ligand docking calculations. For MolFinder calculations, we used the ChEMBL [35] dataset (containing 1,941,412 molecules) to select the initial molecules. ChEMBL was used to increase the diversity of the initial bank than SureChEMBL. The ChEMBL dataset version 27, published in May 2020, was downloaded and used. Among the molecules available in the dataset, 500 randomly extracted molecules were used as the MolFinder input.

### 3.3. Protein-Ligand Docking Energy Calculations

In this study, Werner syndrome helicase (WRN) was used as the target protein [36] (Figure 9). The crystal structure of the protein was downloaded and used from the RCSB Protein Data Bank (PDB) dataset with a PDB ID of 6YHR [37]. We converted the PDB file to the pdbqt format to perform docking calculations using the SMILES of ligands obtained from the SureChEMBL dataset. The openbabel program was used to convert the dataset with the pdbqt format [38]. The partial charges of atoms were calculated using the Gasteiger partial charge calculation method, which is the default option for openbabel [39]. The -h option was used to add hydrogens to the molecule, and the—gen3d option was used to create the 3D structure of the molecule. The Autodock-tools program was used to prepare binding pockets where proteins and ligands interact. Using Autodock tools, we calculated the center of mass where adenosine diphosphate (ADP) (x = −21.277, y = −13.171 and z = 38.710), a native ligand of 6YHR, was attached and set it as the origin. Subsequently, the ADP was deleted, and a 30 Å three-dimensional grid box with the number of x-elements, y-elements, and z-elements of 31 was generated and used as a binding pocket.

AutoDock Vina version 1.1.2 [40] was used for protein-ligand docking. The nrun option, the number of independent docking simulations per molecule, was set to 9. The lowest docking energy was used as the docking energy for that molecule. It required about 12 h with 168 Xeon Gold 6132 cores to complete the docking of 100,000 molecules of the ChEMBL library with Autodock Vina.

To predict the docking scores of newly generated molecules via MolFinder, these data were used as training data for docking energy prediction models. Furthermore, docking calculations of 100 molecules were performed with the best objective function value among the newly generated molecules using MolFinder. They were then compared with the lowest docking energy among 100,000 random molecules from SureChEMBL.

### 3.4. Docking Score Prediction Model

One of the fundamental assumptions of this study is that docking calculations of many molecules can be used to extract information on protein-ligand interactions even when experimental data is insufficient. We started with the SMILES strings of the SureChEMBL dataset as input data and generated pre-computed docking scores as output data. We also converted SMILES into feature vectors using an RDKit to use SMILES in machine learning algorithms for predictive model learning. Feature vectors consist of the RDKfingerprint [41], MACCS key [42], and molecular descriptors.

In this study, SMILES was converted to RDKit fingerprint (2048 bit) and used. In addition to this 2048 RDKit fingerprint, MACCS key-based fingerprints of 166 bits were combined. A total of 17 molecular descriptors were additionally used to consider the overall physical properties of the molecule (Table 1). In summary, a single SMILES string was converted into a 2232-dimensional vector by combining the molecular descriptors with the fingerprint bits.

As the first step in this conversion process, SMILES must be converted to mol-type variables in RDKit to use RDKit modules. Of the 100,000 molecules, 85,432 molecules were used to complete the docking calculations, of which 85,161 were converted to mol types. The docking simulations of 14,568 molecules were not completed, mainly because of their large sizes, which were larger than the grid box size of 30 Å. These molecules were ignored in further calculations.

Thus, 85,161 molecules were used as the final dataset. Of these, 80% (68,129) were used as training data, and 20% (17,032) were used as the test data. The PyTorch [43] deep learning library was used to train the predictive models. The structure of the docking energy prediction model is a multilayer perceptron (MLP) model (Figure 10a).

Before inputting data into the first dense layer, the input vectors were batch-normalized. After batch normalization, the input dimension was put into the first layer, and output was set as 1024 dimensions through the ELU activation function [44]. The output value in the 1024 dimension was used as the input layer of the second layer, and an output vector in 528 dimensions was obtained through the activation function. Finally, 528-dimensional vectors were used as the input layer of the third layer, and the value of the last layer was recorded as the output value. This single scalar value was considered as the predicted docking energy of the molecule. This network structure was trained for 200 epochs, and Adam [45] was used as the optimizer. The learning rate was used as the default value of 0.001, and the dropout rate was 0.3 for the training.

### 3.5. Objective Function

An objective function is a function that quantitatively evaluates the desired properties of a molecule. In this study, the objective function was designed as a linear combination of three properties: the predicted docking score, the QED value [27] indicating the drug-likeness of a molecule, and the similarity to a reference molecule.

The equation of the objective function used in this study is:(1)$$R\left(m\right)=\omega_{D}D\left(m\right)-\omega_{Q}Q\left(m\right)-\omega_{S}S\left(m_{\;};m_{\mathrm{ref}}\right)$$

Equation (1). Purpose function for the molecular design for WRN.

In the above equation, $D\left(m\right)$ is the docking energy obtained from the previously trained docking energy prediction model and indicates the predicted value of the binding energy between the desired target protein and the ligand. $\omega_{D}$ is the weight of $D\left(m\right)$. In this work, the values of $\omega_{D}$ were calculated to be 0.1, 0.05, 0.03, and 0.01. $Q\left(m\right)$ is a quantitative evaluation of the drug-likeness of a given molecule as a QED [27] ranging between 0 and 1; the closer it is to 1, the higher is the drug-likeness. $\omega_{Q}$ is the weight of $Q\left(m\right).$ Finally, similarity *S(m;m_ref_)* was considered to obtain a molecule similar to the reference molecule. $\omega_{S}$ is the weight of $S\left(m;m_{\mathrm{ref}}\right)$. Initially, $\omega_{Q}$ and $\omega_{S}$ are set to one because both terms range from 0 to 1. After obtaining the RDKit fingerprint (2048 bits) of the molecules, the Tanimoto similarity was calculated to calculate this. As the target protein of this study, WRN, has few known inhibitors, we used the information of the inhibitor of BLM [46,47], a very similar protein to WRN (Figure 11 and Appendix A). In other words, we attempted to generate a molecule with high similarity to a molecule that acts as a known inhibitor of BLM. In this study, we used ML216, an inhibitor of BLM helicase, as a reference molecule [46] (Figure 11).

Finally, we also wanted to satisfy Lipinski’s rule of five to generate more drug-like molecules. We penalized the objective function value by adding an arbitrarily large number of 1000 to the predicted docking energy if the molecular weight was greater than 500 Da and the logP was greater than 5. In addition, the radicals were penalized.

### 3.6. Molecular Generation Using MolFinder

This work aims to design drug-like molecules with desirable properties by combining docking score prediction models and the global molecular property optimization approach, MolFinder [2]. MolFinder performs global optimization of a given objective function through combinatorial optimization of SMILES based on the conformational space annealing (CSA) algorithm. The algorithm has the advantage of maintaining the diversity of a set of molecules rather than performing physical optimization based on one molecule, and thus generating more diverse molecules. Previously, we showed that MolFinder generates molecules with better objective function values compared to the existing generative models using reinforcement learning [2]. It should be noted that MolFinder is a general optimization program that finds optimal molecules with any given objective function. In addition, V-Dock can be combined with any optimization method. For example, various RL approaches can be used for V-Dock by using a docking score predictor as a reward function or a part of it.

MolFinder randomly selects a set number of molecules from ChEMBL and stores them in a bank. Here, the molecules with higher objective function values are preferred molecules. The preferred molecules were selected to generate child molecules using crossover and mutation operations. For each preferred molecule, the crossover operation produced 20 child molecules. The mutation operation on 20 child molecules through replacement, removal, and addition, resulted in 80 child molecules for one preferred molecule [2]. Generated child molecules were compared to the bank molecules, and they were compared with the objective functions of the most similar molecules. At this point, the molecules in the bank were updated with molecules having higher objective function values. If the similarity between a molecule and a child molecule in the bank was lower than the average similarity divided by 2, replacement with a molecule having a better objective function value took place after comparing it to the molecule with the worst objective function value in the bank. By iterating these methods, the desired molecules were optimized by updating the bank. Therefore, we set the number of molecules (N_bank_) in the input and output of the molecular bank as 500 and the value of the preferred molecule (N_seed_) as 300. We also iterated the MolFinder calculations 100 times to generate molecules with the highest objective function value.

## 4. Conclusions

This work proposes a computational workflow for generating drug-like molecules by combining the docking energy predictor using SMILES and the efficient global optimization of molecular properties. We used the MolFinder molecular property optimization method to generate molecules that satisfied the desired properties. In this study, we attempted to generate novel molecules that simultaneously satisfied three properties: high docking score, QED, and similarity to a reference molecule. We showed that the average docking energy of the generated 500 novel molecules was −7.81 kcal/mol, which is comparable to that of the known inhibitor, ML216. The mean QED and the mean similarity to ML216 were 0.91 and 0.58, respectively. These results demonstrate that the V-Dock workflow is a highly efficient and novel molecular property optimization scheme. When calculating and comparing the seven properties of QED (molecular weight, logP, number of hydrogen-bonded donor-acceptors, and surface area of molecules) of the generated molecules, we identified that the generated molecules satisfied Lipinski’s rule of five very well. We also showed that V-dock is 255 times faster than actual docking computations and offers an accurate prediction with a correlation coefficient of 0.61. These results demonstrate that using V-Dock enables the fast generation of novel inhibitor candidates for target molecules without sufficient experimental data. As many target molecules do not contain many experimental data, we believe that our V-Dock approach will expand and accelerate the scope of computational drug discovery.

## Figures and Tables

**Figure 1 ijms-22-11635-f001:**
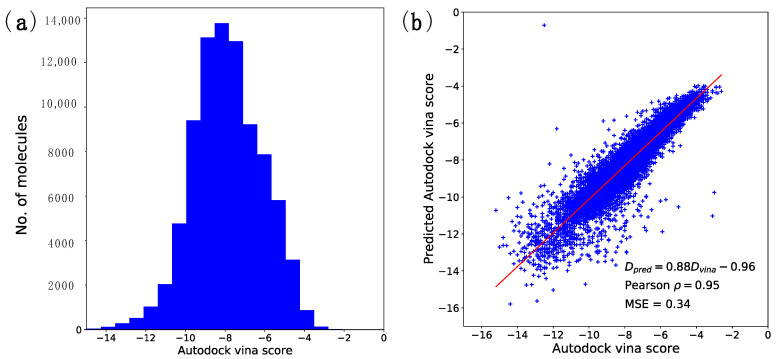
(**a**) Distribution of the docking energies of 100,000 random molecules from ChEMBL. (**b**) A comparison of the true Autodock Vina scores and the predicted scores with the trained predictor. Docking results of 85,161 molecules of the SureChEMBL dataset are represented. Currently, the minimum value of the docking energy is −16.4 kcal/mol, the maximum value is −0.4 kcal/mol, and the average is −7.87 kcal. When the docking calculation result of the SureChEMBL dataset and the docking energy prediction values are compared, the correlation coefficient is 0.95. The mean square error value is 0.34 kcal/mol.

**Figure 2 ijms-22-11635-f002:**
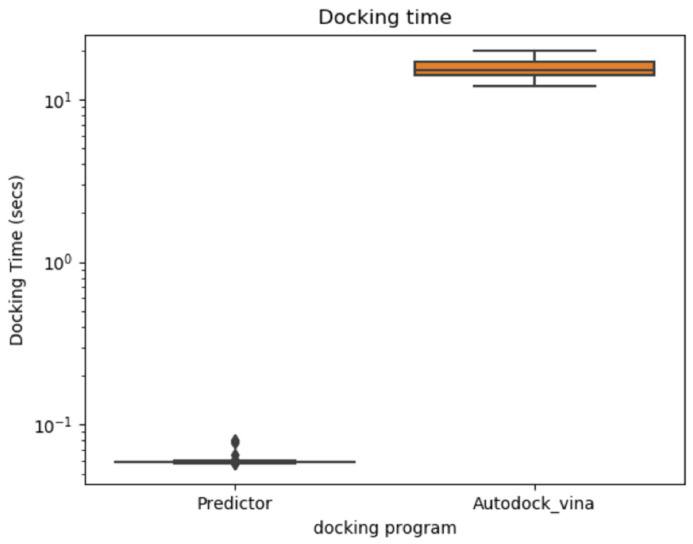
Comparison of computational times of docking energy calculations with Autodock Vina and the prediction model.

**Figure 3 ijms-22-11635-f003:**
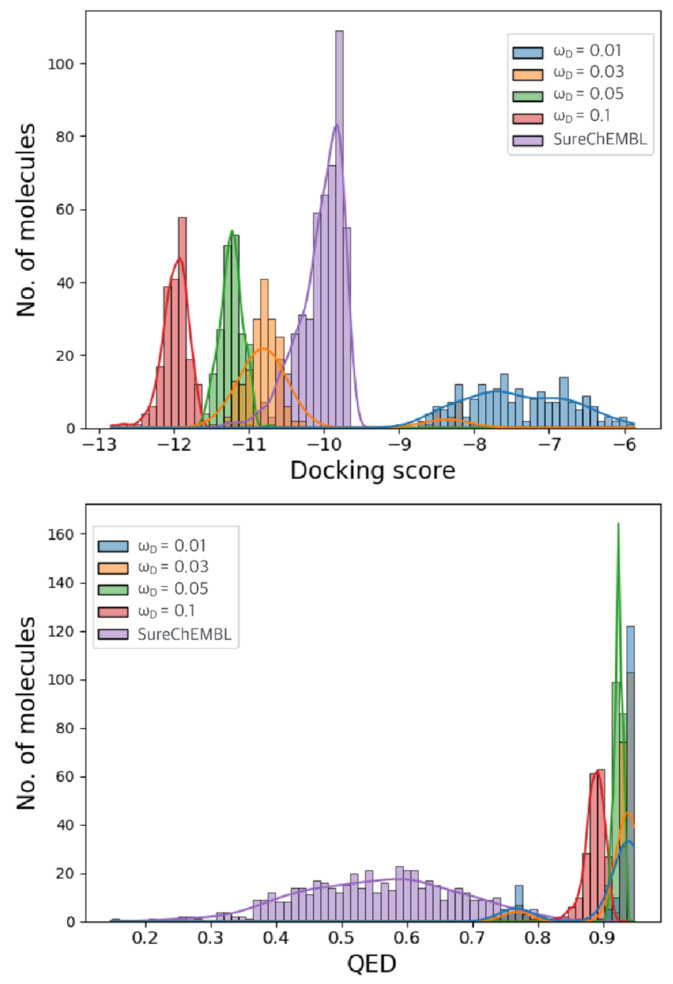
Distributions of the docking score (**top**), QED (**middle**), and a similarity to the reference (**bottom**) of the generated molecules with the known molecules in SureChEMBL while varying the weight coefficient of the docking score term. The distribution of the (**top**) docking energy, (**middle**) QED, and (**bottom**) similarity according to the weight change are plotted. Purple indicates the distribution of the SureChEMBL dataset (the 500 molecules with the lowest docking energy), and blue indicates results when an $\omega_{D}$ value of 0.01 is used. Orange is the property distribution of molecules generated using a $\omega_{D}$ value of 0.03, green indicates the results using a $\omega_{D}$ value of 0.05, and red indicates the results using a $\omega_{D}$ value of 0.1. We can see that the docking energy of the SureChEMBL dataset using the lowest value of $\omega_{D}$ (0.01) is lower than that of the SureChEMBL dataset. Still, QED and similarity show the highest distribution. When using an $\omega_{D}$ value of 0.03, we found that the docking energy of the generated molecule was lower than that of the SureChEMBL dataset value, and the QED generated had higher values than that of the dataset. With $\omega_{D}$ values of 0.1 and 0.05, the molecules with the lowest docking energy were generated using high weights. The QED and the similarity were found to be lower than the values generated using an $\omega_{D}$ value of 0.03.

**Figure 4 ijms-22-11635-f004:**
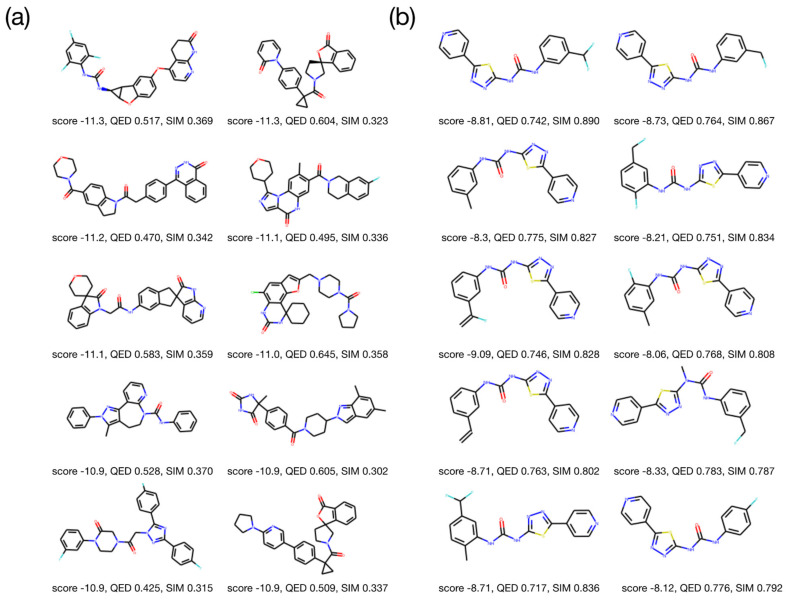
Comparison of existing and generated molecules. (**a**) The ten molecules with the lowest docking energy in the SureChEMBL dataset passed Lipinski’s rule of five. (**b**) The best ten molecules generated by MolFinder. The generated molecules are more similar to ML216 than the known molecules, with high QED and low docking energy values.

**Figure 5 ijms-22-11635-f005:**
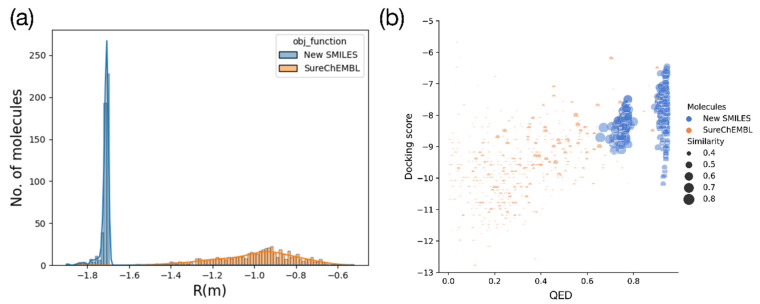
Comparison of (**a**) objective function values in the SureChEMBL dataset with generated molecules, (**b**) the distribution of docking energy, QED, and similarity. The yellow distribution is the SureChEMBL dataset, for 500 molecules with the highest similarity. Blue is a molecule generated using MolFinder. We can see that the generated molecules have a low objective function value distribution between −1.6 and −1.8 and are optimized for the desired properties. The objective function distribution of the SureChEMBL dataset has a value between −0.6 and −1.5, and a higher distribution than the generated molecules. In the figure on the right, the size of the circle denotes similarity, and a larger circle indicates higher similarity to the reference molecule. The *x*-axis is QED, and the *y*-axis is docking energy. The QED of the generated molecule has a much higher value than the SureChEMBL dataset. Since the docking energy is optimized by considering similarity, we can see that it is optimized near −8.5 kcal which is the docking energy of ML216.

**Figure 6 ijms-22-11635-f006:**
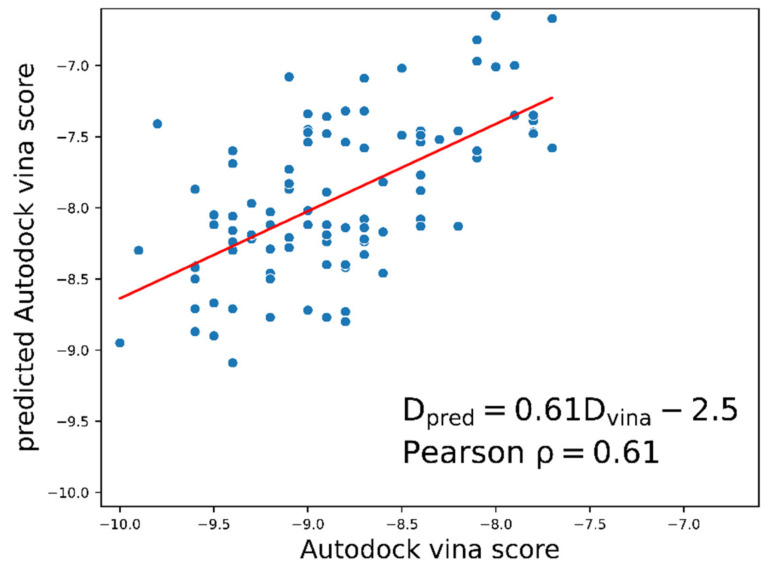
The correlation coefficient between the calculated docking energy and the predicted docking energy. The correlation coefficient between the calculated docking energy and the predicted docking energy is 0.61, using 100 molecules with the lowest objective function value among the generated molecules. This implies that the trained prediction model predicts the dataset and the docking energy of the newly generated molecules accurately.

**Figure 7 ijms-22-11635-f007:**
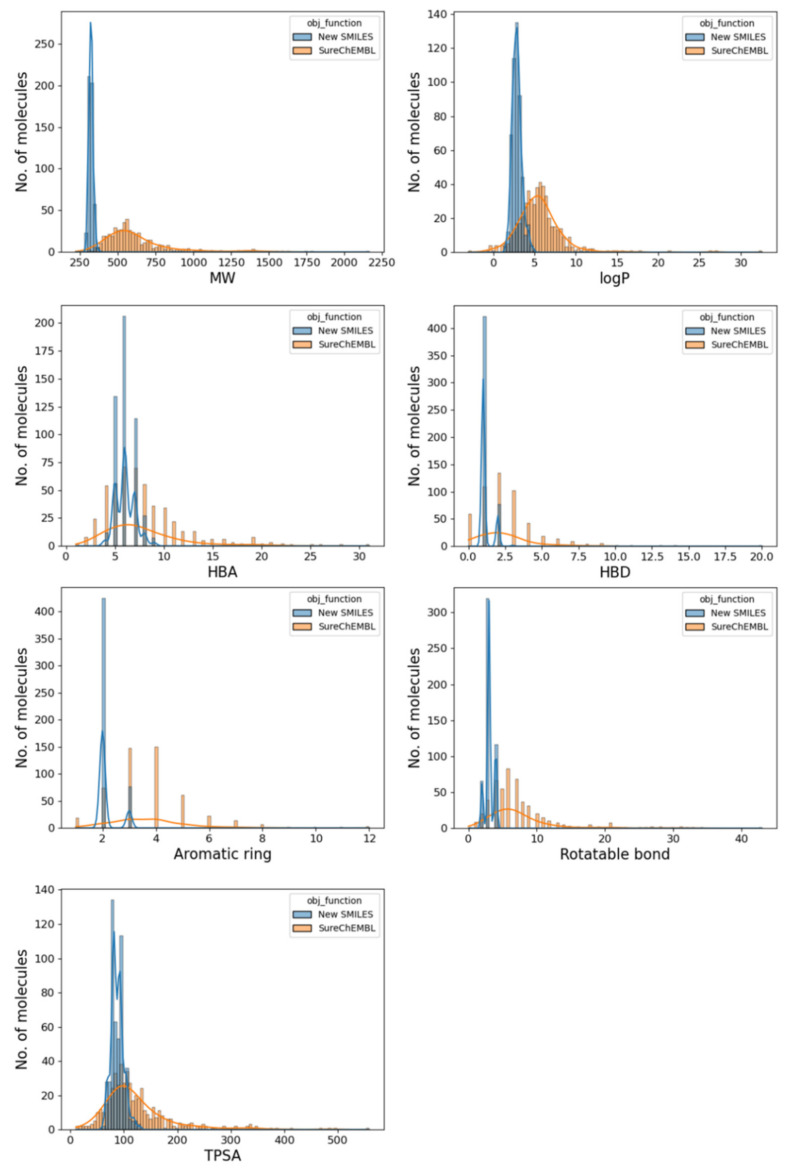
Properties of the 500 molecules produced (molecular weight, logP, number of H-bond donor-acceptors, number of aromatic rings, number of rotatable bonds, and total polar surface area) compared to the corresponding properties of the 500 molecules from SreChEMBL with the highest similarity.

**Figure 8 ijms-22-11635-f008:**
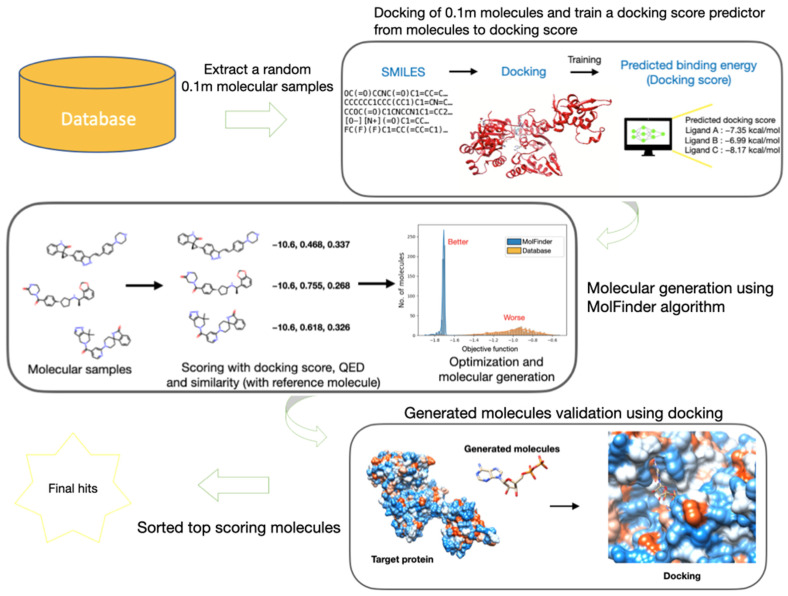
New drug development workflow used in this study.

**Figure 9 ijms-22-11635-f009:**
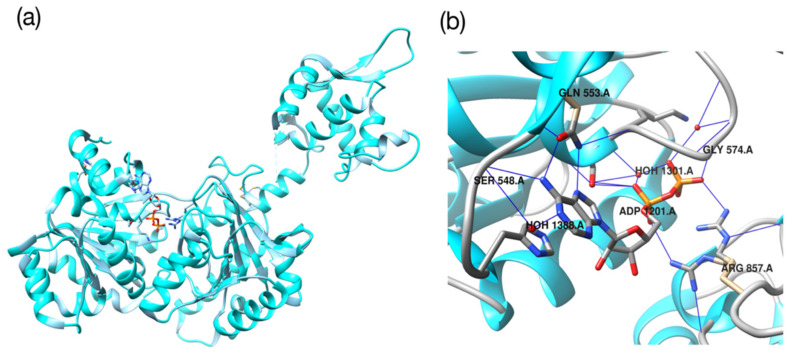
(**a**) Crystal structure of WRN. (**b**) Ligands binding to proteins. Proteins are expressed in the form of blue ribbons and ligands are expressed in the form of sticks. The ligand shown (ADP) is a na-tive ligand of WRN.

**Figure 10 ijms-22-11635-f010:**
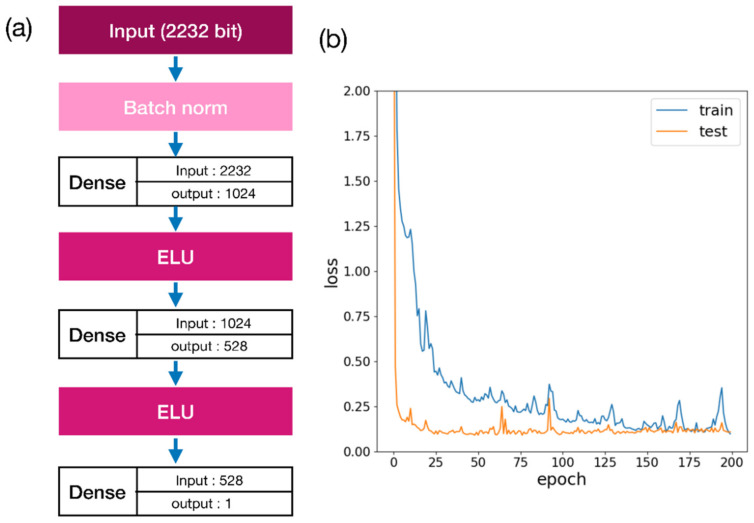
(**a**) Structure of the network of docking energy prediction models. (**b**) Results of training the prediction model. The network’s input used a 2232-dimensional feature vector that combines 17 molecular descriptors, including RDKit fingerprint, MACCS key, and RDKit fingerprint obtained from the molecule’s SMILES. The network output is the predicted docking energy for the Autodock Vina docking energy. In this network, the activation function used the ELU function and Adam as an optimizer. A total of 200 epochs were calculated.

**Figure 11 ijms-22-11635-f011:**
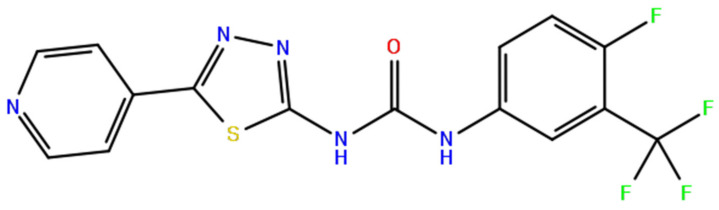
ML216, the known inhibitor of BLM and WRN helicases.

**Table 1 ijms-22-11635-t001:** Types of molecular descriptors.

Feature Name	Describe
HBD	Number of hydrogen bond donors
HBA	Number of hydrogen bond acceptors
Rotatable bond	Number of rotatable bonds
Ring	Number of rings
Radical	Number of radicals
Heteroatoms	Number of hetero atoms
Heterocycles	Number of heterocycles
LipinskiHBA	Number of Lipinski H-bond acceptors
LipinskiHBD	Number of Lipinski H-bond donors
AromaticCarbocycles	Number of Aromatic carbocycles
AromaticHeterocycles	Number of Aromatic heterocycles
AmideBonds	Number of Amide bond
AliphaticCarbocycles	Number of Aliphatic carbocycles
AliphaticHeterocycles	Number of Aliphatic heterocycles
FractionCSP3	Fraction of C atoms that are SP3 hybridized
LabuteASA	Labute Accessible Surface Area
TPSA	Topological polar surface area

## Data Availability

The code and all data are freely available at https://github.com/knu-chem-lcbc/V-dock (accessed on 19 August 2021).

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
