# Peer review of "V-Dock: Fast Generation of Novel Drug-like Molecules Using Machine-Learning-Based Docking Score and Molecular Optimization"

_ijms, 2021, doi:10.3390/ijms222111635_

Round 1

Reviewer 1 Report

This article has an interesting objective, but the methodology and procedure reported is weak, and this manuscript requires several clarifications and careful rewriting is required before it can be considered for resubmission:

  • The purpose of docking programs is to predict a binding pose of the ligand and report energy associated with each binding mode. Docking programs' specially AutoDock Vina have shown great potential in predicting ligand binding mode but not in predicting a correct ranking of the ligand binding (DOI: 10.1371/journal.pone.0155183 and : 10.1021/ci200529n). Authors should address this issue and clearly state that their approach is limited to the performance of AutoDock Vina in the particular system. 
  • The authors stated that out of 85,161 molecules, 80% (68,129) were used as training data and 20% (17,032) test data. Can they show the correlation between vina and V-dock scores for these two classed separately?
  • How does this approach ensure that ligands with bulky sizes that will not fit into the binding pocket (where vina failed to dock!) are penalized in v-dock, and their score will be zero or positive?
  • "We set the nrun option of how many docking calculations per molecule to 9 and used the lowest docking energy as the docking energy for that molecule." What is the nrun option in AutDock Vina?
  • What value of exhaustiveness was used in AutoDock vina since it will affect the average docking time reported in the article substantially. How many processors were used to run docking by vina?
  • All the citations have been changed to "Error! Reference source not found."
  • Last but the most important issue in this manuscript at his point is the English language. It is hard to follow and often not clear what authors want to say. For example, "The protein's crystal structure was downloaded and used on the RCSB Protein Data Bank (PDB) dataset with a PDB ID of 6YHR[36]". It has to be rewritten so that reviewer can understand what the authors want to say.

Reviewer 2 Report

Throughout this manuscript, the authors propose a computational method to design novel drug-like molecules based on three parameters: Docking score, drug-like properties, and similarity with reference molecules. The proposed method uses the calculation of docking energies of a library of compounds to train a machine-learning-based model, which will be subsequently used to predict docking energies straight from the structure (SMILES) of novel molecules generated using MolFinder (just published by the same group).

In my opinion, the manuscript is well written and the proposed method looks very promising. Accordingly, I can recommend the manuscript for its publication. There are however some issues that the authors may address to raise the scope of the manuscript:

Main issues:

  • In my opinion, the weakest point of the manuscript is that it is not fully clear what could be done using just MolFinder, and what is the benefit of combining it with V-dock. For example, using just MolFinder could be possible to generate a series of similar compounds with a reasonable QED?

  • Since the authors are reporting a freely available code, I miss a small section stating the codes that are interfaced with v-dock. Is MolFinder currently a part of V-dock?

  • In section 2.2 the authors state that they used molecules from the SureChEMBL for the docking calculations, while for MolFinder they used ChEMBL. I miss a sentence stating the differences between SureChEMBL and ChEMBL.

  • Please define wD, wQ, and wS in section 2.5.

  • The authors describe thoroughly how wD was chosen, but I could not find references about sensitive values of wQ and wS and how they were chosen.

  • The authors chose wD=0.03 for subsequent calculations due to a compromise between docking score, and similarity to ML216. What is difficult for me is to imagine how different are molecules with a similarity index of 0.7 or 0.4. Perhaps a table showing examples of molecules with different similarities could help to understand the method.

  • Please explain how the seven properties considered for QED were evaluated.

Minor issue:

  • In Fig. 8 b the orientation of the molecules is not conserved. It makes it harder to evaluate the differences between the structures. Besides, the structure of ML216 may be added to better evaluate the structural differences between the molecules. Also in Fig.8, the font size is too small.

Typos:

At least on my computer the figures are not correctly referred to in the text and an error message shows up many times (Error! Reference source not found). Please check

On page 11, the authors discuss results for wD=1.0, however, according to the figure caption wD is 0.1. Please check.
